# Distribution-aware Fairness Learning in Medical Image Segmentation From A Control-Theoretic Perspective

Yujin Oh [* 1]  Pengfei Jin [* 1]  Sangjoon Park [* 2 3]  Sekeun Kim [1]  Siyeop Yoon [1]  Kyungsang Kim [1]
Jin Sung Kim [2 4]  Xiang Li [1]  Quanzheng Li [1]

## Abstract

Ensuring fairness in medical image segmentation is critical due to biases in imbalanced clinical data acquisition caused by demographic attributes (e.g., age, sex, race) and clinical factors (e.g., disease severity). To address these challenges, we introduce Distribution-aware Mixture of Experts (dMoE), inspired by optimal control theory. We provide a comprehensive analysis of its underlying mechanisms and clarify dMoE's role in adapting to heterogeneous distributions in medical image segmentation. Furthermore, we integrate dMoE into multiple network architectures, demonstrating its broad applicability across diverse medical image analysis tasks. By incorporating demographic and clinical factors, dMoE achieves state-of-the-art performance on two 2D benchmark datasets and a 3D in-house dataset. Our results highlight the effectiveness of dMoE in mitigating biases from imbalanced distributions, offering a promising approach to bridging control theory and medical image segmentation within fairness learning paradigms. The source code is available at https://github.com/tvseg/dMoE.

## 1. Introduction

The inherent imbalance and ill-posed nature of data collected in clinical practice, impacted by demographic at-

*Equal contribution  [1]Center for Advanced Medical Computing and Analysis (CAMCA), Department of Radiology, Massachusetts General Hospital (MGH) and Harvard Medical School, MA 02114, USA [2]Department of Radiation Oncology, Yonsei University College of Medicine, Yonsei University, Seoul 03772, Republic of Korea [3]Institute for Innovation in Digital Healthcare, Yonsei University, Seoul 03772, Republic of Korea [4]Oncosoft Inc, Seoul 03776, Republic of Korea. Correspondence to: Xiang Li <xli60@mgh.harvard.edu>, Quanzheng Li <li.quanzheng2@mgh.harvard.edu>.

*Proceedings of the 42nd International Conference on Machine Learning*, Vancouver, Canada. PMLR 267, 2025. Copyright 2025 by the author(s).

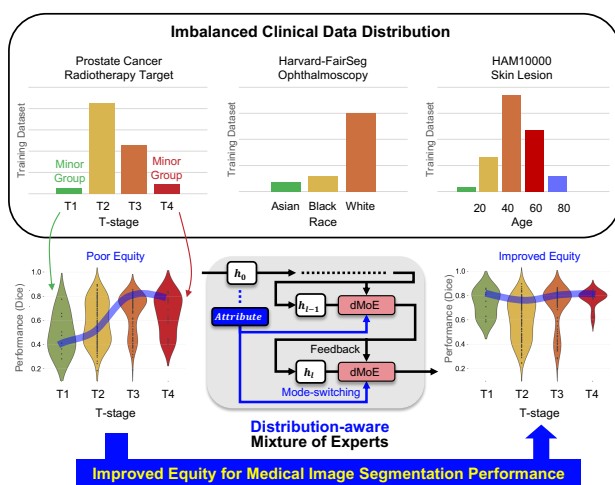

*Figure 1.* The influence of clinical data distribution on medical image segmentation and the role of dMoE as a distribution-aware control mechanism to address inequity challenges. Transparent blue lines within the violin plots connect the most densely populated regions of each attribute, visually representing overall equity.

tributes such as gender and race, as well as clinical factors, e.g., disease severity, emphasize the need for fairness-aware learning approaches in medical data analysis (Ktena et al., 2024; Jin et al., 2024). Deep neural networks, due to their data-driven optimization processes, often overfit to prevalent patterns while failing to adequately learn from underrepresented ones, leading to potentially biased predictions and weakening fairness across demographic subgroups, as shown in Figure 1. Although advanced fairness training strategies have been actively developed (Tian et al., 2025; 2024; Li et al., 2024), distributional approaches remain underexplored—despite the fact that clinical and demographic factors play a significant role in many clinical decision-making (Chester et al., 2020; Theodore et al., 2023). Clinicians consider demographic factors such as age, gender, and race alongside a patient's unique condition to assess both individual and subgroup characteristics. This process not only informs clinical factors like disease stage and decision-making but also implicitly shapes region/population-specific

practice patterns based on cumulative experience and local demographics. However, current fairness learning approaches primarily focus on explicit factors such as demographic attributes but neglect implicit/contextual factors such as disease progression patterns or severity, despite the fact that these factors can be potential influence by region/population characteristics.

Building on these distributional insights and the need for broader applicability across diverse attributes, we propose Distribution-aware Mixture of Experts (dMoE), a framework that adaptively incorporates individual characteristics and distributional patterns into deep neural network training. This architecture is grounded in an in-depth analysis of the Mixture of Experts (MoE) framework (Shazeer et al., 2017) with the adaptation of an optimal control perspective system. By analyzing the structural parallels between MoE and traditional control systems (Åström & Murray, 2021), we reinterpret MoE as a feedback control mechanism. We further enhance its gating mechanism to incorporate distributional information as a mode-switching control for adaptive parameter selection, resulting in dMoE (Figure 2). This approach ensures robust and fair performance across diverse demographic and clinical subgroups.

Moreover, to ensure broad applicability, we integrate dMoE with various network architectures, including transformers (Vaswani et al., 2017) and convolutional neural networks (CNNs), demonstrating the generalizability of its architecture and making the framework well-suited for a wide range of medical image analysis tasks. We validate the effectiveness of dMoE based on the segmentation task on multiple clinical datasets with diverse segmentation masks for diagnosis and treatment planning tasks. Experimental results demonstrate that dMoE not only advances state-of-the-art (SOTA) fairness learning approaches but also presents a way to incorporate distributional attributes to provide robust and equitable diagnosis and clinical decision-making across diverse demographic and clinical attributes.

Our core contributions are summarized as follows:

- We reinterpret MoE from the perspective of feedback control mechanism and extend it into dMoE for fairness learning by incorporating distributional awareness.

- dMoE operates seamlessly within transformers and CNNs, enabling its use in 2D and 3D segmentation tasks across demographic and clinical attributes.

- Extensive medical image segmentation experiments for diagnosis and treatment planning demonstrate dMoE's robustness, demonstrating its effectiveness in mitigating biases from imbalanced medical data distributions.

## 2. Related Work

### 2.1. Fairness Learning in Medical Image Segmentation

Fairness-oriented medical image segmentation datasets with demographic information (Tschandl et al., 2018; Tian et al., 2024) have enabled research on bias mitigation, driving the development of robust fairness training strategies. Advanced generative approaches (Li et al., 2024; Ktena et al., 2024) aim to generate diverse samples from skewed distributions to mitigate bias. However, challenges remain in addressing the computational demands and achieving high-quality sample generation for high-dimensional medical images, such as 3D whole-body computed tomography (CT), for use in augmentation datasets. On the other hand, fairness-focused loss function modifications, such as distributionally robust optimization (DRO) (Sagawa et al., 2019) and fair error-bound scaling (FEBS) (Tian et al., 2024), have been proposed to integrate fairness considerations directly into the optimization process. While partially effective, these methods are vulnerable to the data distribution within the training batch, limiting their applicability in 3D medical image segmentation, where large batch sizes are constrained. Furthermore, existing benchmarks and studies primarily focus on demographic attributes (Tian et al., 2025; Jin et al., 2024), often overlooking critical clinical factors, such as tumor progression and metastasis, which contribute to regional variations in clinical practice patterns due to inherent biases. In this study, we propose a general method to mitigate bias arising from both demographic and clinical factors.

### 2.2. Mixture of Expert in Multi-distribution Learning

Recent advances in the MoE framework (Shazeer et al., 2017) have demonstrated remarkable potential for adapting AI models to diverse data distributions, particularly within continual learning paradigms. MoE enhances robustness and adaptability when confronted with previously unseen data patterns (Rypeść et al., 2024; Yu et al., 2024). In the medical domain, MoE has been effectively extended to address challenges such as multimodal integration (Jiang & Shen, 2024), scanning modality heterogeneity (Zhang et al., 2024), and catastrophic forgetting issues in continual learning (Chen et al., 2024; Wang et al., 2024), unifying these approaches into a cohesive framework that enhances performance. However, theoretical insights into how MoE facilitates the adaptation of disparate distributions to a target distribution remain limited. This study clarifies the underlying mechanism of MoE as dynamic parameter selection, and integrates environmental attributes into its gating mechanism, enabling distributional adaptation for medical image segmentation.

## 2.3. Training Neural Networks as Optimal Control

Neural networks, especially those with shortcut connections, perform complex transformations through successive modifications of a hidden state. These networks can be conceptualized as undergoing a continuous dynamic process, which can be described using ordinary differential equations (ODEs) (Weinan, 2017; Lu et al., 2018). Training these networks resembles solving an optimal control problem, where the objective is to adjust the network parameters, i.e., weights, to minimize a loss function (Chen et al., 2018; Sun et al., 2024).

Fixed-architecture feedforward neural networks can be regarded as operating under a non-feedback control. Employing a consistent strategy, however, can be restrictive in dynamic environments (Åström, 1995). To address this limitation, feedback control mechanisms (Doyle et al., 2013; Åström & Murray, 2021) offer an alternative by enabling continuous monitoring and adjustment based on the system's outputs and desired targets. Furthermore, mode-switching control (Yamaguchi et al., 1996; Boskovic & Mehra, 2000) introduces additional flexibility by enabling the policy to alternate between multiple optimized operational modes in response to external inputs. This capacity is essential for managing complex systems across different conditions and offers a robust response to diverse operational challenges (Yu et al., 2017). Likewise, in the context of image segmentation, adopting varying strategies— such as leveraging distributional attributes of the input—proves beneficial and serves as the motivation for our method.

## 3. Method

In the methods section, we provide theoretical insights and the motivation for proposing the distribution-aware MoE as a solution for fairness learning. We begin by revisiting MoE in Section 3.1 and then expand it into dMoE in Section 3.2. After defining dMoE, in Section 3.3, we further elaborate on the principles of optimal control and demonstrate how dMoE can be formulated as a mode-switching optimal control problem, by providing conceptual connections between them, as illustrated in Figure 2(a) and Figure 2(b) (bottom).

### 3.1. Revisiting MoE

The MoE framework (Shazeer et al., 2017), which serves as the backbone structure for our proposed dMoE, originally leverages sparse gating to achieve computational efficiency while allowing for large model capacity. The output of the MoE layer is defined as follows:

$$y = \sum_{i=1}^{n} G(x)_i E_i(x), \qquad (1)$$

where $n$ is the total number of experts, $E_i(x)$ represents the output of the $i$-th expert network for the input $x$, and $G(x)$ is the output of the gating network, a sparse $n$-dimensional vector that determines which experts are activated. The MoE approach fundamentally relies on sparsity, where the gating network engages only a limited subset of experts for each input, substantially lowering computational overhead by excluding inactive experts from processing. The gating function calculates $G(x)$ using methods such as *Noisy Top-K Gating*, which prioritizes the most relevant $k$ experts while nullifying contributions from the remaining $n - k$. Both the gating network and the experts are optimized simultaneously through backpropagation, ensuring seamless integration and fine-tuning of the MoE layer. This design facilitates a significant expansion of model capacity without proportionally increasing computational demands, making it particularly well-suited for large-scale tasks like language modeling and machine translation.

### 3.2. Distribution-aware MoE

Now, we explain our proposed Distribution-aware Mixture of Experts (dMoE), as illustrated in Figure 2(a). We further provide detailed network architecture in Appendix A.1. Unlike traditional sparse MoE, our proposed dMoE module integrates multiple distribution-wise router networks $G^{attr}$ and a set of $n$ expert modules, consisting of shallow multi-layer perceptron (MLP) neural networks, defined as $E = \{Expert_1, Expert_2, \ldots, Expert_n\}$. We start by flattening intermediate image embeddings from $l$-th layer block output, represented as $h_l \in \mathbb{R}^{H_l W_l D_l \times Ch_l}$, into $\tilde{h}_l \in \mathbb{R}^{N_l \times Ch_l}$, where the total number of embeddings is $N_l = H_l W_l D_l$. Here, $H_l$, $W_l$, $D_l$, and $Ch_l$ correspond to the height, width, depth, and channel dimensions of the intermediate image embeddings, respectively. For the transformer backbone, which already has a flattened dimension, we omit this process. Given these flattend embeddings $\tilde{h}_l$ and an attribute flag $attr$, the activated router $G^{attr}$ identifies the top-$k$ experts. The final output is computed as a weighted sum of the outputs of these selected experts:

$$\bar{h}_l = \tilde{h}_l + \sum_{i=1}^{k} G_i^{attr}(\tilde{h}_l) \cdot E_i(\tilde{h}_l), \qquad (2)$$

where $G^{attr}()$ outputs a weight matrix prioritizing each expert's contribution in a *attr*-specific manner. The resulting weighted output is combined with $\tilde{h}_l$, representing the shared path, to yield the dMoE image embedding $\bar{h}_l \in \mathbb{R}^{N_l \times Ch_l}$. For non-transformer backbones, this representation is reshaped to match spatial dimensions as $\bar{h}_l \in \mathbb{R}^{H_l W_l D_l \times Ch_l}$, where $N_l = H_l W_l D_l$.

In specific, the router network $G^{attr}$ computes sparse weights $H$ using Gaussian noise, as follows:

$$G^{attr}(x) = \text{Softmax}(\text{KeepTop-}k(H(x), k)), \qquad (3)$$

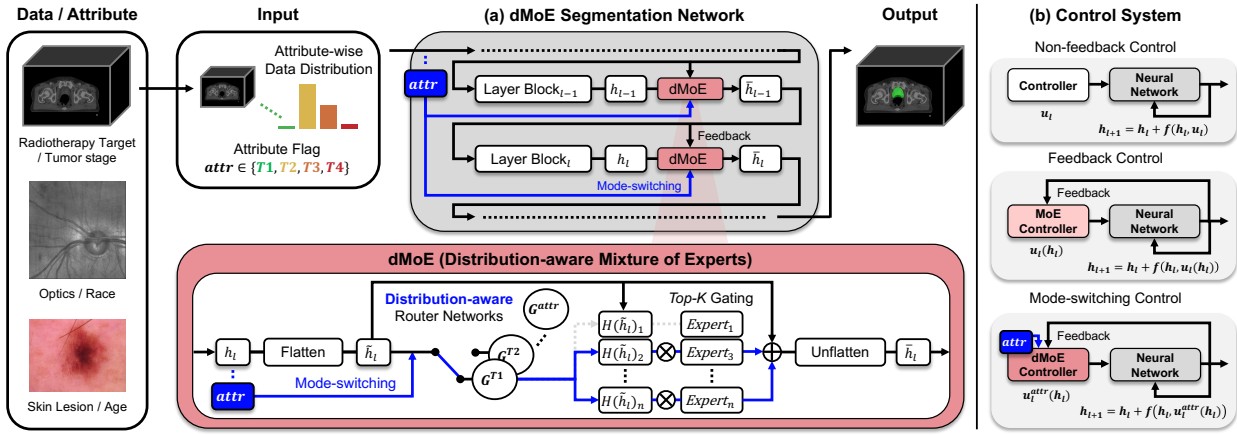

*Figure 2.* (a) Schematic of the dMoE segmentation network for fairness learning, and (b) its interpretation through a control system.

$$H(x)_i = (x^\top \cdot W)_i + \mathcal{N}(0, 1) \cdot \text{Softplus}((x^\top \cdot W^{\text{noise}})_i), \quad (4)$$

$$\text{KeepTop-}k(v, k)_i = \begin{cases} v_i & \text{if } v_i \text{ is in top } k \text{ elements of } v, \\ -\infty & \text{otherwise.} \end{cases} \quad (5)$$

where, Softmax($\cdot$) function normalizes the selected weights, KeepTop-$k(\cdot)$ retrains only the top-$k$ expert contributions, and $W$ and $W^{\text{noise}}$ are trainable weight matrices, and Softplus($\cdot$) is a smooth alternative activation function.

Following the dMoE modules, the final dMoE image embedding $\bar{h}_l$ is passed to the decoder, which predicts the final output, $\hat{y}$. The network is optimized using the segmentation loss:

$$\min_{\mathcal{M}} \mathcal{L}(\hat{y}, y) = -\mathbb{E}_{x \sim P_X} \left[ y_i \log p(\hat{y}_i) \right], \quad (6)$$

where $\mathcal{M}$ represents any 2D-to-3D neural network architecture equipped with our proposed dMoE module, $y \in \mathbb{R}^{HWD}$ is the ground-truth label, and the predicted output $\hat{y} \in \mathbb{R}^{HWD}$, which is computed as follows:

$$\hat{y} = \mathcal{M}(x, attr), \quad (7)$$

where $x$ is the input image and *attr* is the attribute flag.

### 3.3. Interpreting dMoE Through Optimal Control

In this section, we provide a control-theoretic interpretation of deep neural networks by viewing their layer-wise computations as dynamical systems and formulating training as an optimal control problem. This viewpoint offers two key benefits. First, it provides a unified framework for analyzing MoE architectures and yields concrete insights into their adaptive behavior—specifically, how feedback-based control enables more flexible and expressive representations than fixed expert assignments. Second, it facilitates the integration of well-established concepts from control theory

into neural network design, including architectural principles, optimization techniques, and regularization strategies. Building on this interpretation, we further show that dMoE naturally corresponds to mode-switching control—a classical paradigm in control theory—and highlight its potential to promote fairness and improve generalization in deep learning systems.

Neural networks create complex transformations by repeatedly updating their internal states (known as hidden features). A common mathematical description of these transformations is given by:

$$h_{l+1} = h_l + f(h_l, \theta_l). \quad (8)$$

where $h_l$ represents the hidden feature vector at layer $l$, and $f(h_l, \theta_l)$ represents the transformation applied at layer $l$, parameterized by weights $\theta_l$. This iterative updating formula resembles the Euler discretization method used to approximate continuous-time dynamical systems described by ordinary differential equations (ODEs). The continuous-time counterpart of the above discrete transformation can be represented as:

$$\frac{dh_t}{dt} = f(h_t, u_t), \quad (9)$$

where $h_t$ represents the hidden state at continuous time $t$, and $u_t$ denotes a control input or parameters guiding the system dynamics at each instant. To clarify further, the discrete layer index $l$ in the neural network corresponds to specific discrete time points sampled from continuous time $t$. Similarly, parameters $\theta_l$ at discrete layers are analogous to continuous-time control inputs $u_t$, which represent continuously adjustable parameters that influence the evolution of hidden states over time.

In supervised learning applications, the training process of a neural network aims to minimize a loss function $\mathcal{L}$ given

labeled data pairs $(x, y)$:

$$\arg \min_{\{\theta_l\}} \{\mathbb{E}_{\{x,y\}} \mathcal{L}(\hat{y}, y)|$$
$$h_{l+1} = h_l + f(h_l, \theta_l), h_0 = x, \hat{y} = h_L\}. \qquad (10)$$

where $\hat{y}$ is the prediction at the final layer $L$, starting from the input $x$. Here, optimizing over parameters $\theta_l$ corresponds to choosing the best transformations at each discrete step to minimize the discrepancy between predictions and true labels. From the continuous-time viewpoint, this optimization corresponds to a terminal optimal control problem:

$$\arg \min_{\{u^t\}} \{\mathbb{E}_{\{x,y\}} \mathcal{L}(\hat{y}, y)|$$
$$dh_t = f(h_t, u_t), h_0 = x, \hat{y} = h_T\}. \qquad (11)$$

Here, $\hat{y}$ is the system state at the final time $T$ starting from input $x$ and evolving under control inputs $u_t$. Standard feedforward neural networks correspond to Non-feedback control scenarios, meaning the control inputs are determined upfront and do not adapt based on intermediate states. In contrast, the MoE framework introduces adaptive decisions $u_t$ that select transformations based on current hidden states $h_t$—this is precisely a feedback (closed-loop) control mechanism, as illustrated in Figure 2(b) (middle). The corresponding dynamics are given by:

$$dh_t = f(h_t, u_t(h_t))dt. \qquad (12)$$

Given the complexity of optimizing policy function through neural networks, kernel methods (Hofmann et al., 2008) are often utilized to approximate the control function $u_t(h_t)$:

$$u_t(h_t) \approx \sum_i K(h_t, h_t^i) u_t(h_t^i), \qquad (13)$$

where $K$ is a kernel function, $h_t^i$ are predefined anchor points, and $u_t(h_t^i)$ correspond fixed control parameters at these anchor points. This re-parameterization transforms the optimization process from directly optimizing a potentially complex continuous function $u_t$ into optimizing a set of fixed parameters $\theta_t^i = u_t^i(h_t^i)$, significantly simplifying the learning task:

$$f(h_t, u_t(h_t)) \approx f(h_t, \sum_i K(h_t, h_t^i)\theta_t^i). \qquad (14)$$

We consider the following kernel function

$$K(h_t, h_t^i) = \frac{\exp\langle \phi(h_t), \phi(h_t^i) \rangle}{\sum_i \exp\langle \phi(h_t), \phi(h_t^i) \rangle}. \qquad (15)$$

To facilitate the learning of the kernel function $\phi$ we introduce a simplified linear reparameterization. The inner product is computed by: $\langle \phi(h_t), \phi(h_t^i) \rangle = \langle h_t, \phi^* \phi(h_t^i) \rangle = (h_t^\top \cdot W)_i$, where the transformed anchor features $\phi^* \phi(h_t^i)$

form the columns of a learnable parameter matrix $W$, and $\phi^*$ denoting the transpose of $\phi$. Furthermore, for computational efficiency and robustness, we have incorporated Noisy Top-K Gating, corresponding to Eqs. (3), (4), and (5). Specifically, if the function $f$ is linear with respect to its second argument and can be interpreted as the strategy of an expert, where each expert $E_i$ acts as a controller with fixed parameters $\theta_i$, then the following equation holds:

$$f(h_t, \sum_i K(h_t, h_t^i)\theta_t^i) = \sum_i K(h_t, h_t^i)f(h_t, \theta_t^i)$$
$$= \sum_i G_i(h_t)E_i(h_t), \qquad (16)$$

where $G_i(h_t) = K(h_t, h_t^i)$ serve as gating weights dynamically allocating the influence of each expert $E_i$. This formulation naturally leads us to mode-switching control, a classical concept in optimal control that aligns with the design philosophy of dMoE. In this framework, the system selects between multiple control strategies based on both the current state $h_t$ and external contextual parameters $attr$. Concretely, we model this behavior by introducing a switching logic governed by external conditions $attr$. The system dynamics are defined as:

$$dh_t = f(h_t, u_t), \quad u_t = \kappa_{s(attr)}(h_t), \qquad (17)$$

where $\kappa$ denotes a family of control strategies and $s(attr)$ is a switching function that selects the appropriate mode based on environmental attributes. Each mode $\kappa_i$ corresponds to a distinct sub-policy activated under specific conditions. To prevent overfitting and to share knowledge learned from images of different distributions, we utilize a shared expert $E_i$ across different flags $attr$. For each attribute, a distinct $G$ is trained, implying that for different attr, we adopt different functions $\phi$ in Eq. (15), analogous to using different matrices $W$ in Eq. (4):

$$\sum_{i=1}^k G_i^{attr}(h_t) \cdot E_i(h_t). \qquad (18)$$

Considering that the neural network's layerwise structure represents a discrete form of the control leads to Eq. (2). Therefore, dMoE can be interpreted as attribute-wise mode-switching control variant of optimal control for fairness learning, as reflected in the structural resemblance illustrated in Figure 2(a) and (b) (bottom).

## 4. Experimental Results

### 4.1. Datasets

To demonstrate the effectiveness of our proposed dMoE framework as an optimal control approach, we conduct extensive experiments on two benchmark datasets and an

in-house dataset. For each dataset, we visualize the data distribution for each attribute in Figure 1 and provide further details of the training and test datasets in Appendix A.2. For the 2D segmentation experiments, we utilize two datasets: 1) Harvard-FairSeg (Tian et al., 2024) and 2) HAM10000 (Tschandl et al., 2018). For the 3D segmentation experiments with clinical attributes, we utilize our in-house radiotherapy target dataset for prostate cancer patients.

**Harvard-FairSeg** is a scanning laser ophthalmoscopy (SLO) fundus image dataset comprising 10,000 samples with pixel-wise optic cup and outer neuroretinal rim segmentation masks for diagnosing glaucoma. It includes six key demographic attributes—age, gender, race, ethnicity, language preference, and marital status—enabling comprehensive studies of fairness. In this study, we focus on the race attribute, i.e. race $\in$ {Black, Asian, White}, as individuals from minor attribute subgroups face a risk of developing glaucoma compared to other groups, yet the segmentation accuracy is often lowest for this demographic (Tian et al., 2024). Fairness and segmentation performance are evaluated on the test benchmark, which consists of 2,000 samples.

**HAM10000** is a dermatology image dataset comprising 10,015 2D RGB samples with binary segmentation masks for diagnosing skin lesions. It includes demographic attributes such as sex and age, enabling targeted analysis of distributional disparities. In this study, we focus on the age attribute, as younger and older populations are underrepresented in the dataset. For age categorization, patients are divided into four groups at 20-year intervals, with the test benchmark consisting of 1,061 samples.

**Radiotherapy Target Dataset** comprises pelvic CT scans of prostate cancer patients, accompanied by clinical target volume (CTV) segmentation masks for radiotherapy planning in radiation oncology. The dataset includes clinical factors, such as tumor staging and histopathological findings. In this study, we focus on the Tumor (T)-stage attribute, i.e. T-stage $\in$ {T1, T2, T3, T4}, as the dataset exhibits an imbalance in population distribution. For example, radiotherapy can be applied across all T-stages in prostate cancer, but its aim may vary depending on clinical practices. In certain regions, radiotherapy alone is used as definitive treatment for both early and advanced stages without surgery. In contrast, other regions favor radiotherapy alone for early stages, while combining surgery with adjuvant radiotherapy for advanced stages, reflecting institutional variations in treatment patterns. To address biases arising from the imbalanced T-stage distribution when training deep neural networks for radiotherapy target segmentation, we utilize a training dataset comprising 721 primary prostate cancer patients from Yonsei Cancer Center, Seoul, South Korea, and validate network performance using an independent test set,

composed of 132 primary prostate cancer patients from Yongin Severance Hospital, Yongin, South Korea and 143 test samples from Gangnam Severance Hospital, Seoul, South Korea. The data collected for this study has been ethically approved by the IRB of the Department of Radiation Oncology at Yonsei Cancer Center, Yongin Severance Hospital, Gangnam Severance Hospital (IRB numbers 4-2023-0179, 9-2023-0161, and 3-2023-0396).

### 4.2. Implementation Details

For all experiments, we set the dMoE module hyperparameters with Top-$k$ as 2 and the number of experts $n$ as 8. Each expert layer consists of a standard MLP with two linear layers, a ReLU activation, and a dropout layer. For 2D segmentation tasks, we use TransUNet (Chen et al., 2021) as the backbone with the standard ViT-B architecture. The network is trained following the setup used in previous studies. All the input images are center-cropped and resized into 2D patches of size $224 \times 224$ pixels with a batch size of 42. The network is trained with a learning rate of 0.01 for 300 epochs on the Harvard-FairSeg dataset and 100 epochs on HAM10000, following the benchmark setting. The best performance is selected from checkpoints saved at 100-epoch intervals. For 3D radiotherapy target segmentation task, we adopt the 3D Residual U-Net (Çiçek et al., 2016) as the backbone architecture, since it is reported as an effective architecture for radiotherapy target segmentation (Oh et al., 2024). The network is trained using randomly cropped 3D patches of size $384 \times 384 \times 128$ voxels and a batch size of 4. During evaluation, the entire 3D CT volumes are processed using a sliding window approach. The training is conducted with a learning rate of $5 \times 10^{-5}$ over 100 epochs and early stopping based on the validation dataset. The additional computational costs introduced to each backbone are compared in Appendix A.3. We implement the networks using PyTorch (Paszke et al., 2019) in Python with CUDA 11.8 The AdamW (Loshchilov & Hutter, 2017) optimizer with exponential learning rate decay is used for all experiments. For 2D segmentation tasks, we use a single NVIDIA A100 80GB GPU, while for the 3D segmentation task, we use a single NVIDIA RTX A6000 48GB GPU.

### 4.3. Baseline Method and Evaluation Metrics

We evaluate our method against baseline approaches for fairness learning. On the 2D Harvard-FairSeg dataset, we compare our approach with reported metrics from adversarial (ADV) training (Madras et al., 2018), distributionally robust optimization (DRO) (Sagawa et al., 2019), fair-error-bound scaling (FEBS) (Tian et al., 2024), and generative model-based augmentation (FairDiff). We implement our method under the same training conditions as the benchmark methods. For further experiments on the 2D HAM10000 dataset and the 3D radiotherapy target dataset, we compare

our method with FEBS among SOTA methods. We exclude the generative model-based approach, FairDiff, as it requires additional image generation and evaluation tailored to 2D and 3D datasets, which makes it suboptimal for direct comparison. Moreover, we use the default MoE as an additional baseline, replacing our proposed dMoE modules by default MoE modules without distribution-aware router. We implement all methods under the same training conditions as our method.

To evaluate performance, we employ the Dice Similarity Coefficient, Intersection over Union (IoU), as well as equity-scaled segmentation performance (ESSP) metrics for both Dice and IoU—denoted as ES-Dice and ES-IoU—following (Tian et al., 2024):

$$\text{ESSP} = \frac{I(\{(\hat{y}, y)\})}{1 + \Delta}, \tag{19}$$

$$\Delta = \sum_{attr \in A} |I(\{(\hat{y}, y)\}) - I(\{(\hat{y}, a, y) \mid a = attr\})|, \tag{20}$$

where $I \in \{\text{Dice}, \text{IoU}\}$ and $A$ represents the set of all demographic groups. ESSP is fundamentally aligned with both the demographic parity and equalized odds principles (Hardt et al., 2016). While ESSP evaluates performance fairness across diverse demographic subgroups, it has limitations in capturing substantial performance gains for specific subgroups, particularly in the 3D radiotherapy target segmentation task. We complement the limitation of ESSP metrics by providing quantitative analysis with violin plots, which offer a more detailed visualization of equity and performance distribution across diverse subgroups. We also report worst-group accuracy metrics to ensure that the model's lowest performance is properly accounted for.

To support statistical analysis for fairness evaluation, we apply bootstrapped 95% confidence intervals (CIs) when computing both ESSP and all attribute-average metrics, resampling each subgroup and all groups with replacement over 1,000 iterations, respectively.

## 4.4. Results

### 4.4.1. 2D SEGMENTATION WITH DEMOGRAPHIC ATTRIBUTES

We first evaluate dMoE for 2D segmentation on benchmark datasets. For the Harvard-FairSeg neuroretinal rim and optic nerve segmentation tasks, our method consistently achieves SOTA performance in terms of ES-Dice and ES-IoU, surpassing previous fairness learning approaches, as shown in Table 1. Specifically, our method demonstrates superior performance for the minor attribute subgroup (Black), achieving a Dice score of 0.776 for rim segmentation, compared to FairDiff (0.743) and FEBS (0.733). Similarly, for optic cup segmentation, our method significantly improves

performance for the Asian attribute subgroup, achieving a Dice score of 0.844 compared to FairDiff (0.832) and FEBS (0.825). Although MoE, which removes the distribution-aware gating network from our proposed method, achieves comparable performance (0.845), dMoE outperforms MoE across other subgroups for both rim and cup segmentation. Furthermore, the shared structure within dMoE module across subgroups enables synergistic effects, maintaining or even improving performance for the major attribute subgroup (White). For the HAM10000 skin lesion segmentation task, our method also demonstrates SOTA performance in ES-Dice and ES-IoU, with scores of 0.801 and 0.725, respectively, outperforming MoE (0.796 and 0.721) and FEBS (0.757 and 0.667), as shown in Table 2. Specifically, loss function modification methods, such as FEBS, exhibit inferior performance across attribute subgroups compared to the baseline method. This suggests that when the training dataset for each subgroup is insufficient, these methods may sacrifice overall performance. While MoE demonstrates promising performance for minor attribute subgroups, dMoE further improves fairness.

### 4.4.2. 3D RADIOTHERAPY TARGET SEGMENTATION WITH CLINICAL ATTRIBUTES

We further evaluate dMoE for 3D radiotherapy target segmentation, using T-stage as a distributional attribute. Table 3 summarizes the segmentation performance. Despite the test set being acquired from a different hospital with a distinct data distribution, dMoE demonstrates promising generalization performance. The strength of dMoE is particularly evident in underrepresented subgroups, such as T1 and T4 where the frequency is significantly lower than that of T2 and T3, by resulting the most notable performance improvements in these subgroups. Specifically, dMoE demonstrates robust and consistent performance, particularly excelling in the T4 subgroup with a Dice score of 0.778, significantly outperforming FEBS (0.685) and MoE (0.708). Moreover, despite the limited sample size of the T1 subgroup, dMoE achieves superior performance. This consistent improvement demonstrates dMoE's generalizability and robustness, even when applied to datasets from different centers with varying data distributions.

However, as noted in Section 4.3, ESSP metrics have limitations in capturing significant performance gains for minor attribute subgroups. To address this, we further illustrate violin plots in Figure 3(a). Comparison on violin plots, with equity measures indicated by transparent blue lines, confirms dMoE's ability to maintain equity across attribute-wise subgroups. Qualitative results for each subgroup are presented in Figure 3(b). As observed in the ground truth labels, the radiotherapy target volume tends to expand with the progression of the tumor's clinical stage. This trend is distinctly captured by dMoE, which demonstrates superior

*Table 1.* Comparison on 2D Harvard-FairSeg dataset with **race** as the distribution attribute.

| Method | All (n=2000) | | | | Asian (n=169) | | Black (n=299) | | White (n=1532) | |
| --- | --- | --- | --- | --- | --- | --- | --- | --- | --- | --- |
| | ES-Dice (CIs) | Dice (CIs) | ES-IoU (CIs) | IoU (CIs) | Dice | IoU | Dice | IoU | Dice | IoU |
| **Rim Segmentation** | | | | | | | | | | |
| TransUNet[†] (Chen et al., 2021) | 0.703 | 0.793 | 0.585 | 0.671 | 0.746 | 0.616 | 0.731 | 0.599 | 0.811 | 0.691 |
| + ADV[†] (Madras et al., 2018) | 0.700 | 0.791 | 0.583 | 0.668 | 0.741 | 0.612 | 0.729 | 0.598 | 0.809 | 0.689 |
| + DRO[†] (Sagawa et al., 2019) | 0.700 | 0.790 | 0.581 | 0.667 | 0.747 | 0.618 | 0.723 | 0.590 | 0.808 | 0.689 |
| + FEBS[†] (Tian et al., 2024) | 0.705 | 0.795 | 0.587 | 0.673 | 0.748 | 0.619 | 0.733 | 0.602 | 0.813 | 0.694 |
| + FairDiff[‡] (Li et al., 2024) | 0.716 | 0.800 | 0.596 | 0.680 | 0.757 | 0.628 | 0.743 | 0.611 | 0.817 | 0.699 |
| + MoE | 0.733 (0.713-0.752) | 0.804 (0.799-0.809) | 0.614 (0.596-0.633) | 0.685 (0.680-0.691) | 0.760 | 0.635 | 0.763 | 0.635 | 0.817 | 0.701 |
| + dMoE | **0.743 (0.723-0.763)** | **0.813 (0.808-0.818)** | **0.627 (0.608-0.645)** | **0.698 (0.692-0.704)** | **0.769** | **0.645** | **0.776** | **0.652** | **0.825** | **0.713** |
| **Cup Segmentation** | | | | | | | | | | |
| TransUNet[†] (Chen et al., 2021) | 0.828 | 0.848 | 0.730 | 0.753 | 0.827 | 0.728 | 0.849 | 0.758 | 0.850 | 0.755 |
| + ADV[†] (Madras et al., 2018) | 0.826 | 0.841 | 0.727 | 0.743 | 0.825 | 0.726 | 0.842 | 0.748 | 0.843 | 0.744 |
| + DRO[†] (Sagawa et al., 2019) | 0.820 | 0.844 | 0.725 | 0.748 | 0.820 | 0.723 | 0.847 | 0.753 | 0.846 | 0.750 |
| + FEBS[†] (Tian et al., 2024) | 0.825 | 0.846 | 0.727 | 0.750 | 0.825 | 0.725 | 0.848 | 0.755 | 0.848 | 0.751 |
| + FairDiff[‡] (Li et al., 2024) | 0.825 | 0.848 | 0.736 | 0.753 | 0.832 | 0.735 | 0.848 | 0.757 | 0.850 | 0.754 |
| + MoE | 0.830 (0.809-0.847) | 0.854 (0.849-0.860) | 0.739 (0.720-0.754) | 0.762 (0.755-0.768) | **0.845** | **0.757** | 0.842 | 0.748 | 0.857 | 0.765 |
| + dMoE | **0.832 (0.810-0.853)** | **0.862 (0.856-0.867)** | **0.745 (0.722-0.765)** | **0.773 (0.766-0.779)** | 0.844 | 0.755 | **0.851** | **0.761** | **0.866** | **0.777** |

† Metric reported from (Tian et al., 2024). ‡ ES-metrics are recalculated using Eq. (19), based on metrics reported in the original paper (Li et al., 2024), for a fair comparison.

*Table 2.* Comparison on 2D HAM10000 dataset for skin lesion segmentation with **age** as the distribution attribute.

| Method | All (n=1061) | | | | Age ≥ 80 (n=121) | | Age ≥ 60 (n=469) | | Age ≥ 40 (n=328) | | Age ≥ 20 (n=120) | | Age < 20 (n=24) | |
| --- | --- | --- | --- | --- | --- | --- | --- | --- | --- | --- | --- | --- | --- | --- |
| | ES-Dice (CIs) | Dice (CIs) | ES-IoU (CIs) | IoU (CIs) | Dice | IoU | Dice | IoU | Dice | IoU | Dice | IoU | Dice | IoU |
| TransUNet (Chen et al., 2021) | 0.792 (0.737-0.841) | 0.876 (0.863-0.889) | 0.714 (0.664-0.766) | 0.824 (0.809-0.838) | 0.862 | 0.787 | 0.868 | 0.809 | 0.888 | 0.846 | 0.895 | 0.857 | 0.875 | 0.839 |
| + FEBS (Tian et al., 2024) | 0.757 (0.704-0.807) | 0.858 (0.845-0.872) | 0.667 (0.613-0.719) | 0.798 (0.783-0.812) | 0.831 | 0.747 | 0.844 | 0.774 | 0.884 | 0.837 | 0.871 | 0.827 | 0.869 | 0.830 |
| + MoE | 0.796 (0.741-0.844) | 0.882 (0.868-0.895) | 0.721 (0.671-0.770) | 0.833 (0.818-0.846) | **0.864** | **0.794** | 0.875 | 0.820 | 0.889 | **0.851** | **0.904** | **0.869** | **0.882** | **0.850** |
| + dMoE | **0.801 (0.745-0.847)** | **0.884 (0.870-0.896)** | **0.725 (0.673-0.776)** | **0.834 (0.820-0.847)** | **0.864** | 0.791 | **0.881** | **0.824** | **0.890** | 0.850 | 0.901 | 0.866 | 0.880 | 0.846 |

*Table 3.* Comparison on 3D radiotherapy target segmentation with **tumor stage** as the distribution attribute.

| Method | All (n=275) | | | | T1 (n=11) | | T2 (n=129) | | T3 (n=114) | | T4 (n=21) | |
| --- | --- | --- | --- | --- | --- | --- | --- | --- | --- | --- | --- | --- |
| | ES-Dice (CIs) | Dice (CIs) | ES-IoU (CIs) | IoU (CIs) | Dice | IoU | Dice | IoU | Dice | IoU | Dice | IoU |
| 3D ResUNet (Çiçek et al., 2016) | 0.487 (0.447-0.529) | 0.610 (0.589-0.630) | 0.367 (0.336-0.399) | 0.462 (0.440-0.482) | 0.493 | 0.341 | 0.569 | 0.420 | 0.659 | 0.511 | 0.656 | 0.506 |
| + FEBS (Tian et al., 2024) | 0.434 (0.406-0.467) | 0.586 (0.567-0.604) | 0.322 (0.302-0.346) | 0.433 (0.414-0.452) | 0.442 | 0.288 | 0.528 | 0.374 | 0.652 | 0.501 | 0.685 | 0.527 |
| + MoE | 0.452 (0.415-0.492) | 0.608 (0.586-0.628) | 0.342 (0.314-0.372) | 0.461 (0.439-0.482) | 0.492 | 0.345 | 0.542 | 0.393 | 0.674 | 0.532 | 0.708 | 0.557 |
| + dMoE | **0.499 (0.469-0.531)** | **0.650 (0.628-0.671)** | **0.384 (0.358-0.410)** | **0.506 (0.483-0.528)** | **0.718** | **0.571** | **0.585** | **0.435** | 0.693 | 0.556 | **0.778** | **0.641** |

*Note.* The underlined value indicates the worst-group accuracy among distribution attributes for each method.

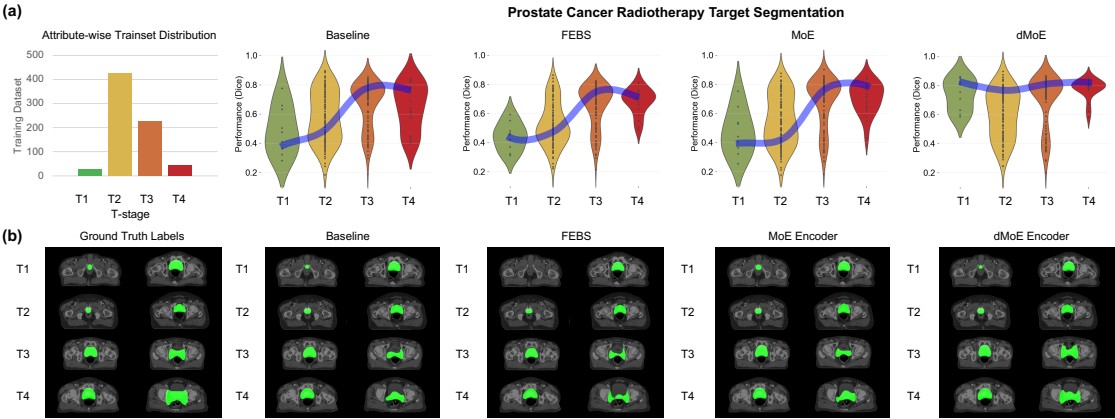

*Figure 3.* (a) Violin plots depicts attribute-wise segmentation performance. Transparent blue lines within the plots connect the most densely populated regions for each attribute, visually representing overall equity. (b) Qualitative comparison across different subgroups.

adaptability to distribution-specific characteristics, thereby achieving notable fairness in performance.

We also present a detailed comparison between the dMoE attribute-wise gating mechanism and multiple independently trained networks in terms of computational complexity and performance, as shown in Table 8 of Appendix A.3. The results demonstrate that dMoE not only achieves higher segmentation accuracy but also maintains greater computational efficiency, outperforming specialized models with a single unified network.

### 4.4.3. RADIOTHERAPY TARGET SEGMENTATION WITH DIVERSE CLINICAL ATTRIBUTES

In clinical practice, radiotherapy targets are determined not solely by tumor stage or visible tumor imaging but through the integration of anatomical imaging with clinical parameters such as T-stage, Gleason Grade Group (GG), and prostate-specific antigen (PSA) levels to account for potential microscopic spread (National Comprehensive Cancer Network, 2024). To evaluate the broader applicability of our method, we further tested additional clinical factors which reflect pathological tumor differentiation but do not indicate the precise tumor location or extent. As shown in Appendix A.4, our results demonstrate fairness improvements comparable to those observed with T-stage. These finding show that incorporating clinical indicators—even when not directly related to visible anatomical structures—can enhance segmentation accuracy across subpopulations.

### 4.4.4. ABLATION STUDIES

We further conduct additional analyses to explore the effects of various configurations of the dMoE module in Appendix A.5. Specifically, we vary the placement of the dMoE module within the encoder and decoder and investigate whether its parameters should be shared or separated across layers to determine the optimal architecture. We further ablate each component of the optimal control framework to assess the contribution of control theory to distribution-aware fairness learning. Detailed results of the ablation study are provided in Table 11 and Table 12.

## 5. Conclusion, Limitations & Future Work

Adopting an optimal control perspective, we design distribution-aware Mixture of Experts (dMoE) architectures to address data imbalance issues. By modeling distribution as an external factor influencing control, we integrate dMoE into medical image segmentation for more effective handling of imbalanced clinical datasets. Given the nature of clinical practice, which accounts for distributional characteristics, our proposed algorithm offers a promising solution to data imbalance and provides a robust framework for clin-ical decision-making that aligns with clinicians' perspectives. Notably, ensuring distributional equity is essential for real-world clinical AI deployment, where training and deployment dataset distributions often differ. Our distribution-aware dMoE holds promise in adapting trained models to unknown distributions, thereby improving the success of clinical AI integration across diverse hospitals.

However, several limitations remain for this study:

First, while we explore diverse attributes across three datasets, the performance improvement trends vary depending on the task and dataset characteristics. Moreover, the optimal configuration of the dMoE module varies between the two different architectures. Future research will focus on uncovering the underlying characteristics of attribute-wise subgroups within each dataset to identify the factors driving these variations and will aim to develop a more generalized module capable of delivering task-agnostic performance improvements.

Second, our focus on a single attribute per task limits the exploration of combined attribute imbalances. For instance, while patients with T2-stage cancer may be prevalent, further subgroup imbalances, such as age or metastasis status, could exist. Addressing this issue may require integrating multiple attributes within a hierarchical dMoE framework, which we plan to explore in future studies.

Last, this study adopts an optimal control perspective to design efficient deep neural network architectures, leveraging back-propagation for parameter optimization. Future research will explore how advanced numerical methods from optimal control theory can inspire novel optimization algorithms specifically tailored to enhance fairness learning.

Despite these limitations, we believe our distribution-aware approach represents a step forward in advancing fairness learning for diverse data-imbalanced clinical scenarios.

## Acknowledgements

This work was supported by the National Institutes of Health, USA R01HL159183, and also supported by the National Research Foundation of Korea (NRF) grant funded by the Korea government (MSIT) (RS-2024-00345854).

## Impact Statement

This paper aims to advance fairness in medical image segmentation by addressing biases arising from imbalanced clinical data distributions. Our work demonstrates robustness across diverse datasets and network architectures, contributing to equitable and reliable AI-driven healthcare applications.

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

# A. Appendix

## A.1. Detailed Network Architecture.

We provide the network architecture for a better understanding of the dMoE modules location within each Transformer-based and CNN-based architecture in Table 4 and Table 5, respectively.

*Table 4.* dMoE within Transformer (TransUNet).

| Module | Layer Block | Resample | dMoE | Data dimension $(C \times H \times W)$ |
|---|---|---|---|---|
| In | - | - | - | $Ch_{in} \times 224 \times 224$ |
| | Conv | - | - | $1 \times 14 \times 14$ |
| Encoder | AttentionBlock$_1$ | - | | $768 \times (14 \times 14)$ |
| | AttentionBlock$_2$ | - | | $768 \times (14 \times 14)$ |
| | $\vdots$ | - | dMoE | $\vdots$ |
| | AttentionBlock$_{11}$ | - | | $768 \times (14 \times 14)$ |
| | AttentionBlock$_{12}$ | - | | $768 \times (14 \times 14)$ |
| Decoder | UpResBlock$_4$ | Up | - | $256 \times 28 \times 28$ |
| | UpResBlock$_3$ | Up | - | $128 \times 56 \times 56$ |
| | UpResBlock$_2$ | Up | - | $64 \times 112 \times 112$ |
| | UpResBlock$_1$ | Up | - | $16 \times 224 \times 224$ |
| Out | Conv | - | - | $Ch_{out} \times 224 \times 224$ |

*Table 5.* dMoE within 3D CNN (3D ResUNet).

| Module | Layer Block | Resample | dMoE | Skip-Connection | Data dimension $(C \times H \times W \times D)$ |
|---|---|---|---|---|---|
| In | Conv | - | - | - | $Ch_{in} \times 384 \times 384 \times 128$ |
| Encoder | ResBlock$_1$ | Down | dMoE$_1$ | ⤴ | $48 \times 192 \times 192 \times 64$ |
| | ResBlock$_2$ | Down | dMoE$_2$ | ⤴ | $48 \times 96 \times 96 \times 32$ |
| | ResBlock$_3$ | Down | dMoE$_3$ | ⤴ | $96 \times 48 \times 48 \times 16$ |
| | ResBlock$_4$ | Down | dMoE$_4$ | ⤴ | $192 \times 24 \times 24 \times 8$ |
| | ResBlock$_5$ | Down | dMoE$_5$ | | $384 \times 12 \times 12 \times 4$ |
| Decoder | UpResBlock$_4$ | Up | - | ↵ | $192 \times 24 \times 24 \times 8$ |
| | UpResBlock$_3$ | Up | - | ↵ | $96 \times 48 \times 48 \times 16$ |
| | UpResBlock$_2$ | Up | - | ↵ | $48 \times 96 \times 96 \times 32$ |
| | UpResBlock$_1$ | Up | - | ↵ | $48 \times 192 \times 192 \times 64$ |
| Out | TransposeConv | Up | - | - | $Ch_{out} \times 384 \times 384 \times 128$ |

## A.2. Dataset Details.

We further provide the trainset and testset for each dataset, along with the attribute subgroup-wise data distribution and percentiles for the trainset in Table 6.

*Table 6.* Detailed distribution of data across attribute subgroups.

| Dataset | Harvard-FairSeg | | | | HAM10000 | | | | | | Radiotherapy Target Dataset | | | | |
|---|---|---|---|---|---|---|---|---|---|---|---|---|---|---|---|
| | Total | Attribute (Race) | | | Total | Attribute (Age) | | | | | Total | Attribute (T-stage) | | | |
| | | Asian | Black | White | | $\geq 80$ | $\geq 60$ | $\geq 40$ | $\geq 20$ | $< 20$ | | T1 | T2 | T3 | T4 |
| Trainset | 7945 | 750 | 1174 | 6021 | 8137 | 191 | 1324 | 3693 | 2356 | 573 | 721 | 26 | 227 | 425 | 43 |
| (%) | (100) | (9) | (15) | (76) | (100) | (2) | (16) | (45) | (31) | (7) | (100) | (4) | (31) | (59) | (6) |
| Testset | 2000 | 169 | 299 | 1532 | 1061 | 121 | 469 | 328 | 120 | 24 | 275 | 11 | 129 | 114 | 21 |

## A.3. Comparison on Computational Efficiency.

We compare the computational complexity of incorporating the dMoE module into the backbone in Table 7. Additionally, Table 8 presents a comparison between the dMoE attribute-wise gating mechanism and multiple independently trained networks, in terms of both computational complexity and performance, focusing on the 3D radiotherapy target segmentation task using tumor stage as the distribution attribute. The results show that dMoE achieves superior performance while maintaining greater computational efficiency, outperforming individually specialized networks with a single unified model.

|  | **TransUNet** | +MoE | +dMoE | **3D ResUNet** | +MoE | +dMoE |
|---|---|---|---|---|---|---|
| Input | 224 W × 224 H | | | 384 W × 384 H × 128 D | | |
| GFlops | 45.84 | 90.28 | 90.28 | 1542.36 | 1761.30 | 1761.30 |
| Params (M) | 91.67 | 129.46 | 129.51 | 13.28 | 26 | 26.05 |

*Table 7.* Computational complexity comparison.

| Method | GFlops ↓ | All (n=275) | | T1 (n=11) | T2 (n=129) | T3 (n=114) | T4 (n=21) |
|---|---|---|---|---|---|---|---|
| | | ES-Dice(D) | Dice | Dice | Dice | Dice | Dice |
| dMoE (Ours) | **1761.30** | **0.499** | **0.650** | **0.718** | **0.585** | **0.693** | **0.778** |
| Multiple networks for each attribute | 5729.44 | 0.457 | 0.606 | 0.599 | 0.515 | 0.681 | 0.760 |

*Table 8.* Comparison to multiple networks for each attribute.

## A.4. Further Clinical Attribute Analysis on Radiotherapy Target Segmentation.

To further evaluate the ability of the proposed dMoE module to capture the diversity of real-world clinical settings, we incorporate an additional clinical parameter as a distribution attribute in the prostate cancer study. Specifically, we use the Gleason Grade Group (GG) and prostate-specific antigen (PSA) level to represent pathological differentiation and disease progression, respectively. As shown in Table 9, our method demonstrates robust performance across various subgroups, particularly in underrepresented groups such as GG 6, 9, and 10. Similarly, Table 10 shows that the proposed dMoE module enhances equity across PSA subgroups, with notable improvements in underrepresented groups such as PSA levels 0 and 4.

*Table 9.* Comparison on 3D radiotherapy target segmentation with **Gleason Grade Groups (GG)** as the distribution attribute.

| Method | All (n=275) | | | | GG level 6 (n=31) | | GG 7 (n=125) | | GG 8 (n=62) | | GG 9 (n=47) | | GG 10 (n=10) | |
|---|---|---|---|---|---|---|---|---|---|---|---|---|---|---|
| | ES-Dice | Dice | ES-IoU | IoU | Dice | IoU | Dice | IoU | Dice | IoU | Dice | IoU | Dice | IoU |
| 3D ResUNet (Çiçek et al., 2016) | **0.512** | 0.610 | **0.389** | 0.461 | 0.562 | 0.417 | **0.578** | **0.429** | 0.650 | 0.501 | 0.669 | 0.517 | 0.623 | 0.474 |
| + FEBS (Tian et al., 2024) | 0.451 | 0.593 | 0.337 | 0.441 | 0.501 | 0.349 | 0.557 | 0.406 | 0.628 | 0.473 | 0.686 | 0.534 | 0.650 | 0.494 |
| + MoE | 0.447 | 0.608 | 0.341 | 0.461 | 0.514 | 0.366 | 0.565 | 0.419 | 0.653 | 0.505 | 0.704 | 0.559 | 0.689 | 0.536 |
| + dMoE | 0.473 | **0.638** | 0.361 | **0.494** | **0.672** | **0.533** | 0.566 | 0.419 | **0.657** | **0.511** | **0.750** | **0.614** | **0.750** | **0.612** |

*Table 10.* Comparison on 3D radiotherapy target segmentation with **prostate-specific antigen (PSA) level** as the distribution attribute.

| Method | All (n=275) | | | | PSA level 0 (n=51) | | level 1 (n=84) | | level 2 (n=60) | | level 3 (n=29) | | level 4 (n=52) | |
|---|---|---|---|---|---|---|---|---|---|---|---|---|---|---|
| | ES-Dice | Dice | ES-IoU | IoU | Dice | IoU | Dice | IoU | Dice | IoU | Dice | IoU | Dice | IoU |
| 3D ResUNet (Çiçek et al., 2016) | 0.504 | 0.608 | 0.380 | 0.460 | 0.552 | 0.402 | 0.578 | 0.429 | **0.643** | **0.498** | 0.643 | 0.495 | 0.659 | 0.508 |
| + FEBS (Tian et al., 2024) | 0.561 | 0.588 | 0.412 | 0.434 | 0.571 | 0.422 | 0.584 | 0.429 | 0.609 | 0.455 | 0.592 | 0.441 | 0.585 | 0.427 |
| + MoE | 0.461 | 0.606 | 0.348 | 0.459 | 0.532 | 0.383 | 0.553 | 0.407 | 0.632 | 0.486 | **0.667** | **0.519** | 0.709 | 0.563 |
| + dMoE | **0.575** | **0.654** | **0.442** | **0.510** | **0.664** | **0.530** | **0.612** | **0.465** | 0.643 | 0.494 | 0.660 | 0.513 | **0.723** | **0.582** |

## A.5. Ablation Study Results.

We perform ablation studies to examine the effects of various configurations of the dMoE module. Specifically, we investigate performance changes based on **1) dMoE Location** within the network, by inserting the dMoE module at different layers within the encoder, the decoder, or both for comparison. We also examine whether **2) dMoE Parameters** should be shared or separated across layers to determine the optimal architecture. In the case of parameter sharing for CNNs, we incorporate additional linear layers before and after the dMoE module to match the channel dimensions across different layer blocks. We further analyze the contribution of **3) Optimal Control Components** by defining the mode-switching feedback control as the dMoE module. To assess its impact, we ablate both the mode-switching and feedback control components. For performing experiments, we use the HAM10000 dataset for Transformer-based architecture (TransUNet), while for CNN-based architecture (3D ResUNet), we utilize a radiotherapy target segmentation task, by reporting performance using the ES-Dice and Dice score as the metric.

As shown in Table 11 and Table 12, inserting the dMoE module within the encoder, decoder, or both yields comparable performance for both architectures, particularly for minority groups (The subgroup with Age < 20 for HAM10000 dataset and the T4-stage subgroup for radiotherapy target dataset). Therefore, we retain the dMoE module within the encoder only to optimize network training efficiency. In the Transformer-based architecture, sharing dMoE parameters across layers enhances performance due to consistent dimensionality between layers. Conversely, in CNN-based architecture, differences in layer-wise dimensionality reduce the effectiveness of parameter sharing. This necessitates adding dimension-matching linear layers before and after the dMoE modules; however, performance remains comparable. As a result, we adopt a layer-wise dMoE configuration specifically tailored for CNNs. In the analysis of optimal control components, a naive adaptation of feedback control resulted in marginal improvements or even performance degradation depending on the dataset. In contrast, our proposed mode-switching control consistently improved both ES-Dice and Dice scores across experiments.

*Table 11.* Ablation study on HAM10000 datsaset for Transformer-based architecture.

| Method | | | All (n=1061) | | Age ≥ 80 (n=121) | Age ≥ 60 (n=469) | Age ≥ 40 (n=328) | Age ≥ 20 (n=120) | Age < 20 (n=24) |
|---|---|---|---|---|---|---|---|---|---|
| | | | ES-Dice | Dice | Dice | Dice | Dice | Dice | Dice |
| | Encoder | Decoder | | | | | | | |
| 1) dMoE Location | ✓(Ours) | | 0.841 | 0.884 | 0.864 | 0.881 | 0.890 | 0.901 | 0.880 |
| | | ✓ | 0.840 | 0.881 | 0.871 | 0.872 | 0.892 | 0.899 | 0.879 |
| | ✓ | ✓ | 0.842 | 0.885 | 0.873 | 0.877 | 0.893 | 0.905 | 0.882 |
| | Layer-Wise | Shared | | | | | | | |
| 2) dMoE Parameters | ✓ | | 0.812 | 0.881 | 0.870 | 0.875 | 0.893 | 0.890 | 0.834 |
| | | ✓(Ours) | 0.841 | 0.884 | 0.864 | 0.881 | 0.890 | 0.901 | 0.880 |
| | Control | Mode-switching | | | | | | | |
| 3) Optimal Control Components | Feedback | ✓(Ours) | 0.841 | 0.884 | 0.864 | 0.881 | 0.890 | 0.901 | 0.880 |
| | Feedback | | 0.836 | 0.882 | 0.864 | 0.875 | 0.889 | 0.904 | 0.882 |
| | Non-feedback | | 0.826 | 0.882 | 0.863 | 0.871 | 0.897 | 0.906 | 0.882 |

*Table 12.* Ablation study on radiotherapy target segmentation for CNN-based architecture.

| Method | | | All (n=275) | | T1 (n=11) | T2 (n=129) | T3 (n=114) | T4 (n=21) |
|---|---|---|---|---|---|---|---|---|
| | | | ES-Dice | Dice | Dice | Dice | Dice | Dice |
| | Encoder | Decoder | | | | | | |
| 1) dMoE Location | ✓(Ours) | | 0.546 | 0.711 | 0.828 | 0.634 | 0.765 | 0.767 |
| | | ✓ | 0.569 | 0.704 | 0.798 | 0.644 | 0.747 | 0.746 |
| | ✓ | ✓ | 0.543 | 0.701 | 0.840 | 0.635 | 0.747 | 0.741 |
| | Layer-Wise | Shared | | | | | | |
| 2) dMoE Parameters | ✓(Ours) | | 0.546 | 0.711 | 0.828 | 0.634 | 0.765 | 0.767 |
| | | ✓ | 0.546 | 0.710 | 0.806 | 0.626 | 0.769 | 0.769 |
| | Control | Mode-switching | | | | | | |
| 3) Optimal Control Components | Feedback | ✓(Ours) | 0.499 | 0.650 | 0.718 | 0.585 | 0.693 | 0.778 |
| | Feedback | | 0.451 | 0.608 | 0.492 | 0.542 | 0.674 | 0.708 |
| | Non-feedback | | 0.509 | 0.615 | 0.524 | 0.573 | 0.668 | 0.637 |

