# OpenReview forum: "Distribution-aware Fairness Learning in Medical Image Segmentation From A Control-Theoretic Perspective"
_ICML.cc/2025/Conference — ICML 2025 spotlightposter_

### Official Review · Reviewer_V3Mq · 2025-03-01

**Overall Recommendation:** 4

**Summary:**

This paper proposes a novel method to address the fairness issue in medical image segmentation by incorporating control theory to handle data distribution disparities. By introducing distribution-aware fairness learning, the method is able to reduce unfairness among different groups while maintaining model performance. Fairness is particularly crucial in the medical field, where the representation of different racial,
gender, and other groups is vital. Therefore, the proposed approach holds significant research value.

**Claims And Evidence:**

Yes

**Essential References Not Discussed:**

No

**Experimental Designs Or Analyses:**

Yes

**Methods And Evaluation Criteria:**

Yes

**Other Comments Or Suggestions:**

See the Other Strengths And Weaknesses

**Other Strengths And Weaknesses:**

1、Strengthsd:
（1）. Innovation in the Use of MOE (Mixture of Experts): The use of MOE to improve fairness learning in the medical image segmentation field is an innovative angle, particularly in addressing the challenge of imbalanced data distributions across different groups.
（2）. Application of Control Theory: Control theory, typically used for optimizing and regulating system behaviors, has been applied here to fairness learning in medical image segmentation, providing a novel perspective.

 2、Weaknesses：
（1）. Insufficient Description of the Patchify Method: The description of the Patchify method (Section 3.2) is rather brief and fails to clearly explain the implementation. There are also formatting and notation errors (such as incorrect subscripts for 'h' in the formula on page 5. It is suggested that the authors provide a more detailed description of the method and correct the formatting issues.
（2）. Lack of Detailed Explanation of the Softplus Function: The Softplus function is mentioned but not explained in detail. Additionally, the content in Section 3.2 seems to be a minor modification of the MOE method, lacking sufficient innovation or explanation.
（3）. Inadequate Derivation of Control Theory Formulation: The derivation of the optimal control theory in Section 3.3 appears abrupt, with insufficient transitions and explanation. The authors are encouraged to provide a more complete mathematical derivation, especially regarding how control theory is integrated with the model, to enhance clarity and persuasiveness.
（4）. Limitations of Experimental Datasets and Design: The experimental datasets are limited in number and relatively small in size. It is recommended that the authors conduct additional experiments on the same dataset using different attribute distributions to further validate the method's effectiveness. Furthermore, expanding the dataset for more comprehensive experiments would be beneficial.
（5）. Insufficient Ablation Studies: The ablation study section is underdeveloped, and it is unclear to what extent the proposed innovations contribute to improvements, particularly regarding the optimal control theory component. The authors should conduct separate ablation experiments to validate the effectiveness of this part.

**Questions For Authors:**

See the Other Strengths And Weaknesses

**Relation To Broader Scientific Literature:**

The paper extends prior frameworks MOE (Mixture of Experts) to address unresolved limitations in fairness learning in the medical image segmentation field

**Theoretical Claims:**

Yes

---

> ### Author Rebuttal · Authors · 2025-04-01
>
> >**R3-W1.1** The Patchify (Section 3.2) fails to clearly explain the implementation.
> - Thanks for your careful comments. Patchify flattens the 4D intermediate image embeddings ‘h’ in [H, W, Z, Ch] from CNN blocks to 2D flattened embeddings ‘\tilde{h}’ in [N, Ch], where N corresponds to H x W x Z, ensuring compatibility of the embeddings with the transformer-based dMoE module. After ‘tilde{h}’ going through the gating x expert network, it needs to be reverted to 4D shape to be further computed with consecutive CNN blocks. This is only necessary for CNN-based architecture, as highlighted in line 134-right, however, we will clarify it in Section 3.2 as well as Figure 2.
>
> >**R3-W1.2** There are also formatting and notation errors (such as incorrect subscripts for 'h' in the formula on page 5).
> - The potential ambiguity of the subscript 'h' might stem from the distinction between 'l' and 't'. In this context, we follow the work on NeuralODE (Section 2.3) which establishes a connection between discrete neural network (NN) structures and continuous dynamic processes. Therefore, in the NN structures and parameters, 'l' is used as the subscript. In discussions of the dynamic process, it serves as the subscript. Moreover, 'l' is treated as a discretization of 't', as seen in E.q 10 and 11. We will provide a more detailed derivation and clarify notations to eliminate any potential misunderstandings.
>
> >**R3-W2.1** Lack of detailed explanation of the Softplus.
> - We will update the explanation: “Softplus is a smooth alternative activation function to ReLU, which is defined as SoftPlus(x) = log(1+e^x)”.
>
> >**R3-W2.2** Section 3.2 seems to be a minor modification of the MOE, lacking sufficient innovation or explanation.
> - Our innovation, grounded in an in-depth analysis of MoE, bridges MoE and optimal control theory and leverages well-established mode-switching control theory for fairness learning. While our modification is simple, it is insightful (based on above analysis) and significant, which addresses a critical unmet need in clinical AI: mitigating performance degradation caused by unbalanced data distribution.
>
> >**R3-W3**  The derivation of the optimal control theory in Section 3.3 appears abrupt, with insufficient transitions and explanation. The authors are encouraged to provide a more complete mathematical derivation, especially regarding how control theory is integrated with the model, to enhance clarity and persuasiveness.
> - We appreciate your constructive point and would like to clarify that our formulation aligns with related work that interprets neural networks through the lens of dynamic processes (Section 2.3). However, we will clarify some points that might have caused ambiguity and add preliminary background and detailed derivations in the final version: (1) The distinction between 'l' and 't' has been explained in R3-W1.2. (2) In our model, 'u' represents the control input: In non-feedback control, it corresponds to NN parameters (\theta), while in feedback control, 'u' becomes a function of 'h', forming an ensemble of experts dependent on 'h' through a kernel method.
>
> >**R3-W4** Conduct additional experiments on the same dataset using different attribute distributions to further validate the method's effectiveness. Furthermore, expanding the dataset for more comprehensive experiments would be beneficial.
> - Due to the character limit, we kindly refer the reviewer to our responses in Resp_Tables 2-4 for additional experimental results using different attributes and Resp_Table 1 for expanded test dataset, both provided in response to R2-W1 of Reviewer qTb4.
>
> >**R3-W5** The ablation study section is underdeveloped. The authors should conduct separate ablation experiments to validate the effectiveness of the optimal control theory component and what extent the proposed innovations contribute.
> - We performed an ablation study on Optimal Control components for radiotherapy target segmentation, as shown in Resp_Table 6.
>
> ***Resp_Table 6.*** Ablation study on Optimal Control components.
> |Methods|Optimal Control|All (n=275)|T1 (n=11)|T2 (n=129)|T3 (n=114)|T3 (n=21)|
> |-|-|-|-|-|-|-|
> |||ES-Dice(D)/D|D|D|D|D|D|
> |dMoE (Ours)|Mode-switching Feedback|0.499/**0.650**|**0.718**|**0.585**|**0.693**|**0.778**|
> |(a)|Feedback|0.451/0.608|0.492|0.542|0.674|0.708|
> |(b)|Non-feedback|**0.509**/0.615|0.524|0.573|0.668|0.637|
>
> - To further evaluate our innovation, we compare dMoE’s attribute-wise gating mechanism with multiple networks trained separately for each attribute. As shown in Resp_Table 7, dMoE demonstrates superior performance and computational efficiency.
>
> ***Resp_Table 7.*** Comparison to multiple networks for each individual attribute.
> |Methods||All (n=275)|T1 (n=11)|T2 (n=129)|T3 (n=114)|T3 (n=21)|
> |-|-|-|-|-|-|-|
> ||GFlops↓|ES-Dice(D)/D|D|D|D|D|D|
> |dMoE (Ours)|**1761.30**|**0.499**/**0.650**|**0.718**|**0.585**|**0.693**|**0.778**|
> |Multiple networks for each attribute|5729.44|0.457/0.606|0.599|0.515|0.681|0.760|

---

> > ### Comment · Reviewer_V3Mq · 2025-04-04
> >
> > All my concern about the work has been  addressed.

---

> > > ### Author Response · Authors · 2025-04-04
> > >
> > > We appreciate the reviewer for raising the score and we are glad that our rebuttal effectively addressed your valuable concerns. Following your suggestions, we will further clarify our method’s effectiveness and clinical significance in the manuscript.

---

### Official Review · Reviewer_qTb4 · 2025-03-10

**Overall Recommendation:** 3

**Summary:**

The paper "Distribution-aware Fairness Learning in Medical Image Segmentation From A Control-Theoretic Perspective" explores the issue of fairness in medical image segmentation, particularly in cases where demographic and clinical factors contribute to biased model performance. The authors argue that biases in deep learning models arise due to the inherent imbalance in clinical data acquisition, often skewed along dimensions such as age, sex, race, and disease severity. As discussed in the introduction, existing fairness-aware training strategies mainly focus on demographic attributes while overlooking clinical factors that influence medical decision-making. This paper proposes a novel approach, termed Distribution-aware Mixture of Experts (dMoE), which adapts deep learning models to heterogeneous distributions in medical imaging.

The study builds upon the Mixture of Experts (MoE) framework. The authors reinterpret MoE as a feedback control mechanism (Section 3.3), where distributional attributes are incorporated into the gating function. This enables the model to adaptively select experts based on demographic and clinical contexts, thereby improving fairness in segmentation tasks. Unlike previous fairness-learning methods, dMoE integrates distributional awareness at the architectural level rather than as a post-hoc correction. As shown in Equations (1)–(5), the dMoE gating mechanism operates using a mode-switching control paradigm, dynamically selecting the optimal expert networks based on subgroup attributes.

The authors validate their approach through experiments on three medical imaging datasets: Harvard-FairSeg for ophthalmology, HAM10000 for skin lesion segmentation, and a 3D radiotherapy target dataset for prostate cancer segmentation. The results, summarized in Tables 1–3, show that dMoE outperforms baseline methods in terms of fairness and segmentation accuracy across underrepresented subgroups.

**Claims And Evidence:**

The claims made in the submission are generally well-supported. The claim that dMoE improves fairness in medical image segmentation is strongly supported by quantitative results in Tables 1–3, which compare its performance against existing fairness-learning approaches. The experiments consistently demonstrate that dMoE achieves state-of-the-art performance, particularly for underrepresented subgroups such as Black patients in ophthalmology and older patients in dermatology. Furthermore, Figure 3 illustrates how dMoE achieves a more balanced segmentation performance across demographic and clinical attributes. The study provides some evidence for generalization in the 3D segmentation experiment, where the test set is sourced from a different hospital than the training data. However, this dataset remains relatively small, with only 132 test cases. To substantiate this claim more convincingly, further external validation across multiple independent datasets would be necessary.

**Essential References Not Discussed:**

I am not aware of missing references.

**Experimental Designs Or Analyses:**

The experimental design and analyses in the paper appear to be overall sound. The experimental design includes tests on three distinct medical imaging datasets. This selection allows for an evaluation of fairness across both demographic attributes (race, age) and clinical attributes (tumor stage). The inclusion of a 3D dataset is a strength of the study. However, the sample sizes of the test datasets are relatively small, particularly for the prostate cancer segmentation task (n=132 test cases).

**Methods And Evaluation Criteria:**

The proposed methods and evaluation criteria in the paper are generally well-aligned with the problem of fairness in medical image segmentation. The choice of using a distribution-aware mixture of experts (dMoE) as an architectural modification is sensible for this problem. By integrating demographic and clinical attributes into the expert selection process, dMoE offers a more structured approach to fairness that dynamically adjusts to subgroup-specific biases. The fairness evaluation metric, equity-scaled segmentation performance (ESSP), is a suitable to qunatify performance.

**Other Comments Or Suggestions:**

No further comments.

**Other Strengths And Weaknesses:**

To the best of my knowledge, the integration of MoE with fairness-aware medical image segmentation appears to be novel. Unlike traditional MoE models, which select experts based solely on feature space partitioning, the proposed algorithm introduces attribute-aware gating functions (Equation 3) that enable the network to adjust its expert selection dynamically based on fairness-sensitive factors.

One of the main weaknesses is the limited external validation and dataset diversity. While the paper evaluates dMoE on three medical imaging datasets, these datasets may not fully capture the diversity of real-world clinical settings. The prostate cancer test dataset in particular is relatively small, with only 132 test samples. Another limitation is the lack of statistical significance testing in the fairness evaluations.

**Questions For Authors:**

Could you clarify how ESSP relates to common fairness metrics in Machine Learning, such as demographic parity, equalized odds, and worst-group accuracy?

**Relation To Broader Scientific Literature:**

To the best of my knowledge, most previous applications of MoE in medical imaging have focused on multimodal learning (Jiang & Shen, 2024) and heterogeneous scanning modalities (Zhang et al., 2024), rather than fairness. The novelty of dMoE lies in integrating fairness as a gating criterion, which has not been explicitly explored in prior MoE applications. However, I cannot be more specific, since I am not very familiar with the literature pertaining to this work.

**Theoretical Claims:**

There are no proofs in this paper.

---

> ### Author Rebuttal · Authors · 2025-04-01
>
> >**R2-W1** Limited external validation and dataset diversity, which may not fully capture the diversity of real-world clinical settings. The prostate cancer test set in particular is relatively small, with only 132 test samples.
> - Thanks for your constructive feedback. For the prostate cancer test set, we collected an additional 143 test samples from a different hospital, more than doubling the total test sample size. These samples were scanned using a different CT manufacturer (SIEMENS) than the training data (Canon). As shown in Resp_Table 1, dMoE demonstrated the most promising and robust fairness performance on the expanded dataset.
>
> ***Resp_Table 1.*** Radiotherapy target segmentation with **tumor stage** on the expanded testset.
> |Methods|All (n=275)|T1 (n=11)|T2 (n=129)|T3 (n=114)|T3 (n=21)|
> |-|-|-|-|-|-|
> ||ES-Dice(D)/D|D|D|D|D|D|
> |RedUNet|0.487/0.610|0.493|0.569|0.659|0.656|
> |+ FEBS|0.432/0.590|0.442|0.528|0.652|0.685|
> |+ MoE	|0.451/0.608|0.492|0.542|0.674|0.708|
> |+ dMoE|**0.499/0.650**|**0.718**|**0.585**|**0.693**|**0.778**|
>
> - Moreover, our used three datasets do not fully capture the diversity of real-world clinical settings. However, datasets containing fairness-related attributes for segmentation remain scarce [1]. Therefore, for the existing dataset, we incorporated another clinical parameter. For prostate cancer, we used Gleason Grade Group (GG), which reflects pathological differentiation and impacts both patient distribution and radiotherapy target patterns. As shown in Resp_Table 2, our method demonstrated robust performance across different subgroups, particularly in underrepresented subgroups such as GG 6, 9, and 10.
> - **Reference**: [1] Yu Tian et al., ICLR 2024, https://arxiv.org/abs/2311.02189
>
> ***Resp_Table 2.*** Radiotherapy target segmentation with **Gleason Grade Groups (GG)**.
> |Methods|All (n=275)|GG 6 (n=31)|GG 7 (n=125)|GG 8 (n=62)|GG 9 (n=47)|GG 10 (n=10)|
> |-|-|-|-|-|-|-|
> ||ES-Dice(D)/D|D|D|D|D|D|
> |RedUNet|**0.512**/0.610|0.562|**0.578**|0.650|0.669|0.623|
> |+ FEBS|0.451/0.593|0.501|0.557|0.628|0.686|0.650|
> |+ MoE|0.447/0.608|0.514|0.565|0.653|0.704|0.689|
> |+ dMoE|0.473/**0.638**|**0.672**|0.566|**0.657**|**0.750**|**0.750**|
>
> - Additionally, we incorporated the gender attribute for the other datasets in Resp_Tables 3 and 4. However, due to the relatively balanced distribution and the absence of established evidence indicating an effect of gender attribute on segmentation patterns, the performance gains of dMoE were less pronounced. Nevertheless, dMoE outperformed other methods for Harvard-FairSeg while showing comparable performance to MoE for HAM10000.
>
> ***Resp_Table 3.*** Harvard-FairSeg dataset with **gender**.
> |Methods|All (n=2000)|Female (n=1229)|Male (n=771)|
> |-|-|-|-|
> ||ES-Dice(D)/D|D|D|
> |TransUNet|0.844/0.848|0.851|0.846|
> |+ FEBS|0.846/0.849|0.851|0.0.849|
> |+ MoE|0.845/0.854|0.850|**0.860**|
> |+ dMoE|**0.856/0.858**|**0.857**|0.859|
>
> ***Resp_Table 4.*** HAM10000 dataset with **gender**.
> |Methods|All (n=1061)|Female (n=496)|Male (n=566)|
> |-|-|-|-|
> ||ES-Dice(D)/D|D|D|
> |TransUNet|0.862/0.879|0.890|0.869|
> |+ FEBS|0.846/0.860|0.869|0.853|
> |+ MoE|**0.880**/0.882|0.881|**0.882**|
> |+ dMoE|0.871/**0.883**|**0.891**|0.877|
>
> >**R2-W2** The lack of statistical significance in the fairness evaluations.
> - To address the lack of statistical analysis in fairness evaluation, we have adapted Bootstrapping Confidence Intervals (CIs) when calculating ESSP metrics, by resampling each subgroup sample with replacement for 1,000 iterations. We will update all the metrics with the 95% CIs as exemplified in Resp_Table 5.
>
> ***Resp_Table 5.*** Statistical significance analyzed Dice metric and an additional  Worst-group accuracy metric for radiotherapy target segmentation.
> |Metric|ES-Dice (CIs)|Dice (CIs)|Worst-group Accuracy|
> |-|-|-|-|
> |RedUNet|0.487 (0.447-0.529)|0.610 (0.589-0.630)|0.493|
> |+ FEBS|0.434 (0.406-0.467)|0.586 (0.567-0.604)|0.438|
> |+ MoE|0.452 (0.415-0.492)|0.608 (0.586-0.628)|0.492|
> |+ dMoE|**0.499 (0.469-0.531)**|**0.650 (0.628-0.671)**|**0.585**|
>
> >**R2-W3** Could you clarify how ESSP relates to common fairness metrics, such as demographic parity, equalized odds, and worst-group accuracy?
> - Thanks for your informative comment. We will explain the ESSP metric in detail in the appendix of the final version. (1) Demographic parity ensures that the probability of a positive outcome is the same for all demographic groups, and (2) Equalized odds ensures false positive and false negative rates to be equal across demographic groups. ESSP aligns with both (1,2) principles by maintaining comprehensive segmentation performance, such as Dice or IoU, across different groups. Whereas, (3) Worst-group accuracy ensures the model's lowest performance is still adequately addressed. ESSP emphasizes equity across worst- to best-group performance, which does not fully align with worst-group accuracy. Therefore, we will include Worst-group accuracy in the main paper, as exemplified in Resp_Table 5.

---

### Official Review · Reviewer_Fj34 · 2025-03-13

**Overall Recommendation:** 3

**Summary:**

The paper proposes a distribution-aware image segmentation framework inspired by the control theory in mode switching and closed loop control. The framework incorporates the mixture of expert to address the heterogeneous distributions in medical images. Experiments in two 2D image benchmarks and a 3D in-house dataset shows superior performance in mitigating the bias.
Thanks for the detailed response. The topic is promising, and the interpretation of the method is sound. The rebuttal is complete, and most concerns are well addressed. Thus, I raise the score to weak accept given the update after rebuttal.

**Claims And Evidence:**

1.	The paper claims the framework to be distribution aware. But section 3.2 doesn’t highlight how is the proposed dMoE more distribution aware compared to a normal MoE: a normal MoE also has an input dependent gate [equation 1] in choosing the experts and thus also “distribution-aware”. The “Distribution-wise router” for each attribute seem to leverage fine-grained attribute annotations of the dataset, which means the user knows in advance which group the input belongs to. This reduces the method contribution of “distribution-aware” in the network as users are required to be aware of it first.
2.	There is no explanation on the design expressed in equation 4: why incorporate two different learnable matrices W and W_noise?
3.	The paper claims to get inspiration from the optimal control theory in the framework design. But the description related to the control theory is confusing and miss explanations on many used notations, making the contribution of the control theory to the proposed framework hard to follow.

**Essential References Not Discussed:**

No.

**Ethical Review Concerns:**

NA.

**Experimental Designs Or Analyses:**

I checked experiments in section 4.4. In table 2 and table 3, dMoE do not outperform MoE in all subgroups. Since the manually chosen attribute wise router in dMoE is actually overfitting to each subgroup’s images, this result is a bit surprising. Further discussions on these results would be valuable.

**Methods And Evaluation Criteria:**

The paper criticizes that “current fairness learning approaches primarily focus on explicit factors such as demographic attributes but neglect implicit/contextual factors such as disease progression patterns or severity” [line 47-50, right column]. However, experiments conducted in dataset Harvard-Fairseg only considers race (a demopraphic attribute), HAM10000 only considers age (a demographic attribute), and Radiotherapy Target Dataset only considers tumor stage attribute (a clinical factor indicating severity). Thus the experiments cannot justify additional benefit of the proposed approach in considering both demographic and implicit/contextual factors.

**Other Comments Or Suggestions:**

Maybe motivating the framework to be more computationally efficient than training multiple networks in each individual attribute would be beneficial for the writing.

**Other Strengths And Weaknesses:**

Strengths: It could be potentially a good application approach when attribute annotations are available as a unified framework handling different types of biases individually.
Weakness: The section on interpreting dMoE through optimal control doesn’t seem to bring additional insights.

**Questions For Authors:**

1.In what scenario will a user know the severity of a tumor in advance and only wants to segment the tumor area? Typically, when users know the severity of the tumor, the user already knows the tumor area. This setting seems not very likely to happen in real world. I may change my mind if reasonable scenarios are described.

**Relation To Broader Scientific Literature:**

The paper is broadly related to fairness-learning [1] where balancing strategies are reviewed and control theory [2] where mode switching is used based on the control signal.
[1] Xu, Zikang, et al. "Addressing fairness issues in deep learning-based medical image analysis: a systematic review." npj Digital Medicine 7.1 (2024): 286.
[2] Yamaguchi, T., Shishida, K., Tohyama, S., and Hirai, H. Mode switching control design with initial value compensation and its application to head positioning control on magnetic disk drives. IEEE Transactions on Industrial Electronics, 43(1):65–73, 1996.

**Theoretical Claims:**

Many notations are not explained in the first time they are used and some notations are never explained. The confusing description makes it hard to build the connection between the control theory and the proposed framework.
[line129-131, right column]: shape of the h and \tilde{h} are the same?
[line 145, right column] why does the patched embedding, after going through gating network*expert network, suddenly become unpatched?
[line 156] Normal() is a weird notation. Is this a real value distributed in gaussian distribution?
[equation 8] what is \theta? What is f?
[equation 9, 12] what is u? t?
[line 205, right column] what is u_i^i(h_t^i) ?
[line 208, right column] what is the shape of \theta?
[line 240] what is the system parameter? Manual setting?

---

> ### Author Rebuttal · Authors · 2025-04-01
>
> >**R1-W1.1** How is dMoE more distribution aware compared to a normal MoE?
> - Thanks for the reviewer’s insightful point. The improved "distribution-awareness" of dMoE stems from its subgroup-aware gating, whereas a normal MoE employs per-sample gating. This design enables each gating to better capture shared patterns within subgroups, enabling subdistribution-aware gating.
>
> >**R1-W1.2/Q1** In what scenarios would a user know the tumor severity in advance and only need segmentation? Typically, when users know the severity of the tumor, the user already knows the tumor area. I may change my mind if reasonable scenarios are described.
> - Thanks for the reviewer's thoughtful comment, and we appreciate the chance to clarify the clinical rationale behind our study design. While it may seem intuitive that knowing tumor severity implies knowledge of tumor location and radiotherapy (RT) target, this assumption does not fully align with standard RT practice. In clinical settings, RT target delineation occurs after initial tumor diagnosis and is not determined solely by tumor severity nor visible tumor imaging. Instead, it integrates anatomical imaging and clinical parameters, such as T stage and Gleason Grade Group (GG), to account for potential microscopic spread.
> - For instance, in prostate cancer, for early stage (T1–T2), even when tumors appear localized on imaging, the entire prostate gland is typically included due to potential microscopic disease beyond visible boundaries. For advanced stage (T3–T4), RT volumes expand further to cover extracapsular extension or adjacent organ invasion such as bladder or rectum, with possible elective nodal irradiation depending on clinical risk.
> - Additionally, clinical factors such as GG represent pathological tumor differentiation but do not specify exact tumor location or extent. Our experiment by subgrouping with GG (Resp_Tables 2 in response to R2-W1) demonstrated fairness improvements, similar to T stage. This emphasizes that integrating clinical indicators—even if not directly related to visible structures—enhances segmentation accuracy by identifying shared visual features within severity groups.
>
> >**R1-S1** Highlighting the framework's computational efficiency compared to training multiple networks for each attribute would be beneficial.
> -  Due to the character limit, we kindly refer the reviewer to Resp_Table 7 in response to R3-W5, which demonstrates dMoE's efficiency.
>
> >**R1-W2** Why incorporate W and W_noise?
> - W_noise further imposes a controlled level of randomness into the expert selection process into the gating mechanism, for avoiding convergence to a few dominant experts.
>
> >**R1-W3** The description for control theory is confusing.
> -  Please see inline-answers bellow.
>
> > [line129-131, right] shape of the h and \tilde{h} are same? [line 145, right] why does the patched embedding, after going through gating*expert network, become unpatched?
> - Please refer to R3-W1.1.
>
> > [line 156] Normal() is a weird notation.
> - It should be updated to N(0,1), the standard normal distribution.
>
> > [E.q 8] What is f, \theta [line 208, right] the shape of \theta? [E.q 9, 12] What is u? t?
> - Please refer to R3-W1.2 and R3-W3.
>
> > [line 205, right] What is u_i^i(h_t^i)?
> - It should be u_t(h_t^i), where h_t^i​ is the i-th anchor point, and u_t(h_t^i) is the value taken at h_t^i​.
>
> > [line 240] What is the system parameter? manual?
> - Instead of manually designing a system, we interpret the neural networks (NN)'s operations on hidden states as a dynamical system. That said, these NN parameters serve as the system parameters, governing hidden state dynamics in accordance with dynamical systems theory.
>
> >**R1-W4** The experiments cannot justify the additional benefit in considering both demographic and contextual factors.
> - dMoE is designed to adapt to each dataset by effectively incorporating attributes that influence performance and distribution, unlike traditional methods that are working on specific datasets. However, as noted in Section 5, we plan to integrate both demographic and contextual factors in future studies.
>
> >**R1-W5** dMoE does not outperform MoE in all subgroups.
> - As discussed in R1-W1.1, MoE, as a distribution-aware mechanism, can outperform in major subgroups. However, the performance gains dMoE provides for minor groups are particularly valuable, and the strength of dMoE lies in its ability to promote balanced performance gains.
>
> >**R1-W6** Interpreting dMoE via optimal control doesn’t bring additional insights.
> - Interpreting NNs through the lens of dynamic processes (Section 2.3) is an active research area with both theoretical and practical benefits: (1) It enhances understanding of the mechanisms behind NN, such as why a MoE outperforms a fixed experts-based NN. (2) It enables the transfer of well-established concepts from control theory to NNs, including architectural structures, optimization, and regularization techniques such as mode-switching mechanisms.

---

> > ### Comment · Reviewer_Fj34 · 2025-04-07
> >
> > Thanks for the detailed response. The topic is promising, and the interpretation of the method is sound. The rebuttal is complete, and most concerns are well addressed. However, some questions could benefit from further interpretation, for example, they explained that dMoE uses "subgroup-aware gating" while normal MoE uses "per-sample gating," but didn't fully clarify the technical distinction or substantiate why this makes dMoE inherently more distribution-aware. Despite these minor points, I raise the score to weak accept.

---

> > > ### Author Response · Authors · 2025-04-07
> > >
> > > We thank the reviewer for recognizing the soundness and promise of our topic, as well as for the improved score.
> > > To further clarify the technical distinction between the standard MoE and our proposed dMoE, we will elaborate on the comparison between conventional optimal control and mode-switching control in Section 3.3 - “Interpreting dMoE Through Optimal Control” of the main paper, thereby enhancing the interpretation of more distribution-awareness of dMoE.

---

### Decision · Program_Chairs · 2025-05-01

**Decision:**

Accept (spotlight poster)

**Comment:**

This manuscript proposes a Distribution-aware Mixture of Experts (dMoE) approach for fairness-aware medical image segmentation, reinterpreting the MoE framework through control theory. The work demonstrates how incorporating demographic and clinical attributes into expert selection can improve segmentation fairness across underrepresented groups. The reviewers highlighted several key strengths, including the novel integration of control theory principles, comprehensive evaluation across multiple medical imaging domains, and demonstrated improvements in fairness metrics. The authors effectively addressed initial concerns by expanding their test dataset for prostate cancer segmentation, incorporating additional clinical attributes like Gleason Grade Groups, and providing statistical validation through bootstrapped confidence intervals. While reviewers initially expressed concerns about limited dataset sizes and theoretical foundations, the authors' detailed rebuttal, including expanded experiments and clarified mathematical derivations, largely satisfied these concerns. The discussion phase showed evolving reviewer perspectives, with Reviewer Fj34 raising their score to weak accept despite some remaining questions about the technical distinction between dMoE and standard MoE, while Reviewer V3Mq maintained a strong accept after their concerns were addressed. Reviewer qTb4 acknowledged the authors' thorough response regarding statistical validation and expanded dataset analysis. The consensus among reviewers improved following the rebuttal, with particular appreciation for the practical clinical relevance and theoretical grounding of the approach.